# Tropical volcanoes synchronize eastern Canada with Northern Hemisphere millennial temperature variability

Feng Wang [1,5] ✉, Dominique Arseneault [1], Étienne Boucher[2], Fabio Gennaretti [3], Shulong Yu[4] & Tongwen Zhang[4]

Although global and Northern Hemisphere temperature reconstructions are coherent with climate model simulations over the last millennium, reconstructed temperatures tend to diverge from simulations at smaller spatial scales. Yet, it remains unclear to what extent these regional peculiarities reflect region-specific internal climate variability or inadequate proxy coverage and quality. Here, we present a high-quality, millennial-long summer temperature reconstruction for northeastern North America, based on maximum latewood density, the most temperature-sensitive tree-ring proxy. Our reconstruction shows that a large majority (31 out of 44) of the coldest extremes can be attributed to explosive volcanic eruptions, with more persistent cooling following large tropical than extratropical events. These forced climate variations synchronize regional summer temperatures with hemispheric reconstructions and simulations at the multidecadal time scale. Our study highlights that tropical volcanism is the major driver of multidecadal temperature variations across spatial scales.

Millennial temperatures reconstructed from climate proxies provide crucial historical insights into the temporal and spatial variability of the Earth's climate, as well as benchmarks for quantifying the recent warming and evaluating the realism of climate model simulations[1–4]. Global and Northern Hemisphere temperature reconstructions now depict high coherence with simulations at the multidecadal time scale, enhancing the predictability of the climate system[2,5]. In contrast, at subcontinental scales, preindustrial temperature variations seem to be region-specific[6] and less spatially coherent in reconstructions than simulations[7,8]. However, it remains unclear to what extent such regional differences reflect high internal variability of the climate system or inadequate proxy coverage and quality[8–10], especially in regions where high-quality proxy data are lacking.

Northeastern North America (hereafter the NENA region) is an archetypal example of this situation where regional uncertainties are large. It is one of the regions of the Northern Hemisphere that lacks millennial-long maximum latewood density (MXD) data, the most sensitive tree-ring proxy for reconstructing summer temperature variability with annual resolution[5,11–13]. Only 12 MXD chronologies have so far allowed reconstruction of summer temperatures back to 1000 CE, with 11 of them clustered in Eurasia (Supplementary Table 1), leading to a well-known gap in MXD-based temperature reconstruction in North America[13–15] (Fig. 1a).

In this study, we fill this gap and develop a highly replicated, temperature-sensitive MXD network from four sites along a 900 km latitudinal transect. Based on this dataset, we show that an important

[1]Département de Biologie, Chimie et Géographie, Centre d'Études Nordiques, Université du Québec à Rimouski, Rimouski, QC G5L 3A1, Canada. [2]Département de Géographie, GEOTOP, and Centre d'Études Nordiques, Université du Québec à Montréal, Montréal, QC H2X 3R9, Canada. [3]Institut de Recherche sur les Forêts, Groupe de Recherche en Écologie de la MRC-Abitibi, Centre d'Étude de la Forêt, Université du Québec en Abitibi-Témiscamingue, Amos, QC J9T 2L8, Canada. [4]Xinjiang Key Laboratory of Tree-Ring Ecology, Key Laboratory of Tree-Ring Physical and Chemical Research, Institute of Desert Meteorology, China Meteorological Administration, 830002 Urumqi, China. [5]Present address: Centre Eau Terre Environnement, Institut National de la Recherche Scientifique, Québec, QC G1K 9A9, Canada. ✉e-mail: feng.wang@uqar.ca

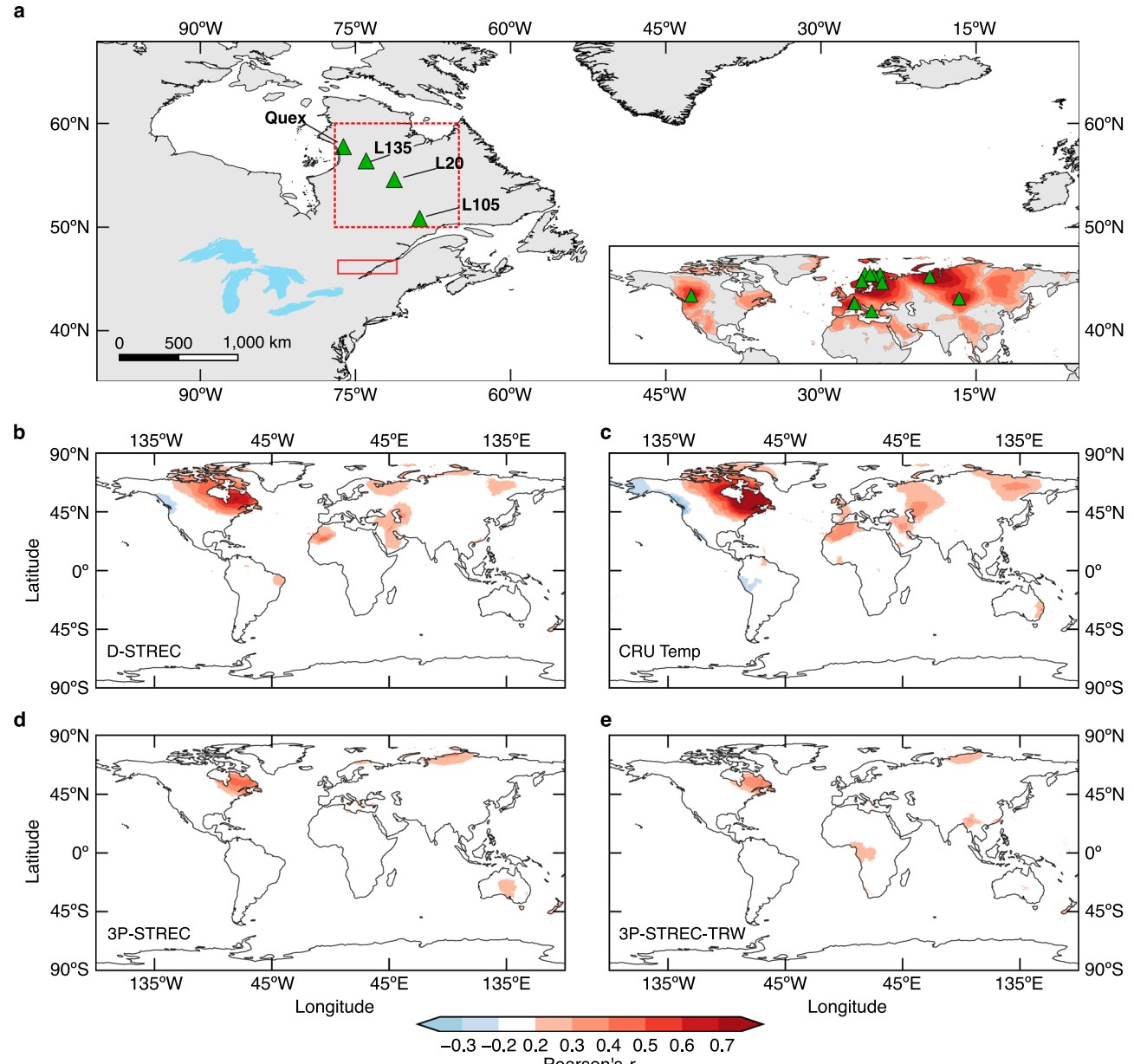

**Fig. 1 | Millennial maximum latewood density (MXD) network and correlation fields with gridded May–August (MJJA) temperatures of the CRU dataset.**
**a** Locations of the MXD network in northeastern North America (NENA) with an inset showing the existing millennial MXD chronologies previously used for summer temperature reconstructions across the Northern Hemisphere (see details in Supplementary Table 1) as well as their correlations with MJJA temperatures over the 1901–1976 common time interval. The inset map shows the maximal positive $r$-values ($P < 0.05$) across correlation fields of individual MXD series, thus indicating the North American data gap. The red dotted square (50°–60°N, 65°–77°W) encloses the area used to calculate the regional temperature target from the CRU dataset. The red solid square refers to the area where the southern Quebec historical temperatures were recorded. **b–e** Spatial domains of NENA reconstructed and observed (CRU) MJJA temperatures over the 1905–2006 period. 3P-STREC is the multiproxy reconstruction[19] discussed in the main text and 3P-STREC-TRW is the tree-ring-width component included in 3P-STREC (Methods). All field correlations are based on 30-year high-pass filtered time series (Butterworth filter) to avoid influence of the long-term warming trend, and only significant correlations ($P < 0.05$ accounting for autocorrelation in time series) are shown. Geographic borderlines are made with Natural Earth. Free vector and raster map data @ naturalearthdata.com.

proportion of the multidecadal temperature variability in NENA is externally forced by explosive volcanism, especially tropical eruptions, synchronizing regional variations with those of the whole Northern Hemisphere.

## Results and discussion
### Reconstruction robustness
We performed a large number of MXD measurements from 1249 black spruce (*Picea mariana* (Mill.) B.S.P.) lake subfossil logs and adjacent lakeshore living trees to build three well-replicated millennial

chronologies (Supplementary Fig. 1) in the boreal forest of NENA (Fig. 1a; Supplementary Table 2). The existing but shorter and less replicated Quex dataset (1363–1989 CE; 45 trees) was added to our network (Fig. 1a; Supplementary Method 1). Overall, this large and unique dataset is robust, with a yearly replication ≥15 trees and an expressed population signal[16] (EPS) > 0.85 since 772 CE (Fig. 2a; Supplementary Fig. 2).

We then developed an MXD-based summer (May–August, MJJA) temperature reconstruction for the 772–2017 period (hereafter D-STREC; Fig. 2b; see Methods). Correlation with the MJJA temperature

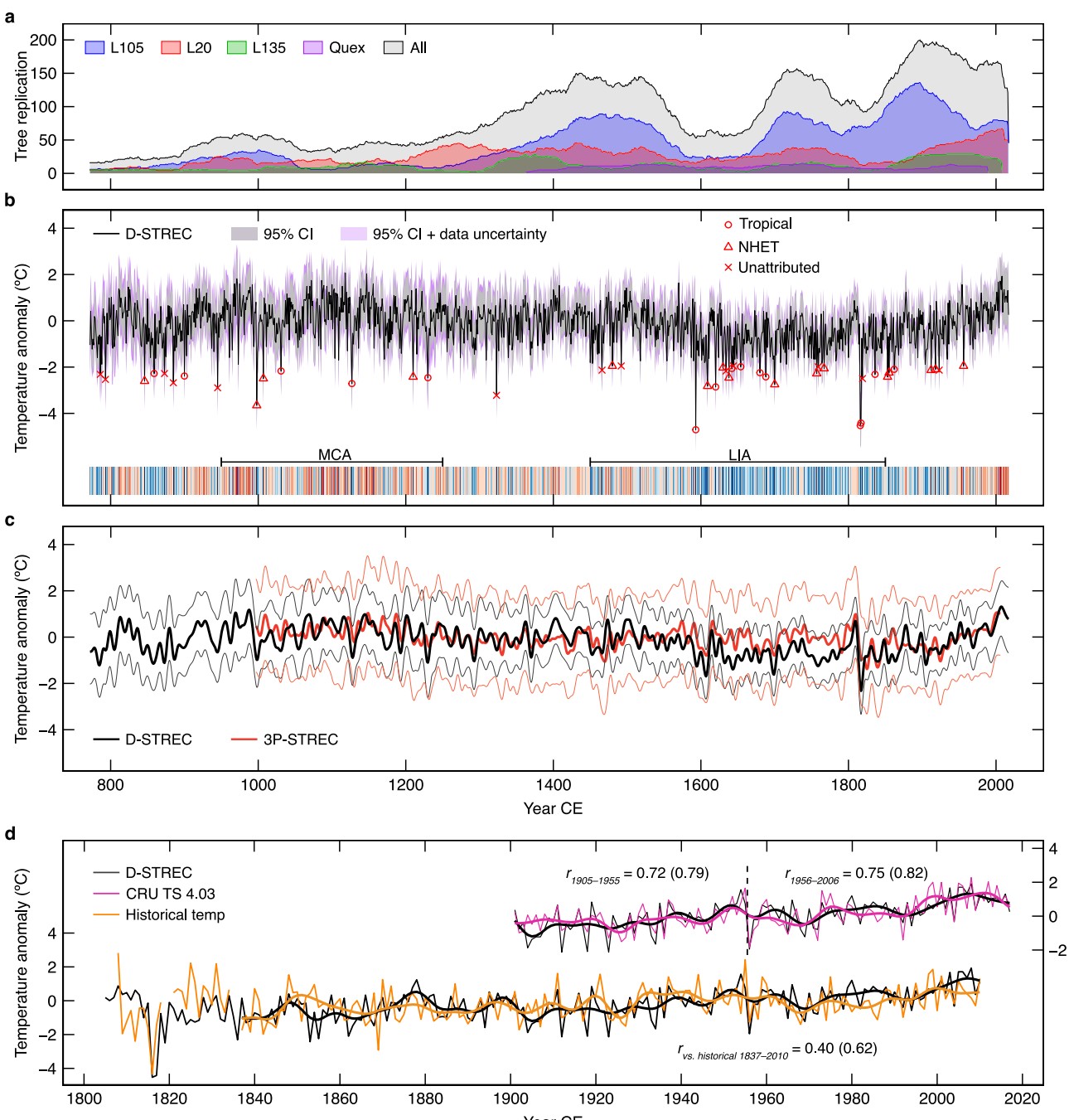

**Fig. 2 | Reconstructed May–August (MJJA) temperature anomalies with respect to 1905–2006 CE in northeastern North America. a** Replication of maximum latewood density series on tree basis. **b** D-STREC (772–2017 CE) superimposed on its two 95% confidence intervals (CIs), along with the temperature stripe graphic (expressed in 10 colors from cold-blue to warm-red) and the Medieval Climate Anomaly (MCA; ~950–1250 CE) and the Little Ice Age (LIA; ~1450–1850 CE) defined by the Fifth Assessment Report of the Intergovernmental Panel on Climate Change[22]. Circles, triangles, and crosses denote cold extremes (≤mean−2 SD) matched to tropical and Northern Hemisphere extratropical (NHET) volcanic

eruptions, and the unattributed extremes, respectively (Supplementary Table 6). The "95% CI" is the uncertainty of the Bayesian model, while the "95% CI + data uncertainty" additionally considers the proxy-level uncertainty (Methods). **c** Comparison of 10-year low-pass filtered (Butterworth filter) D-STREC and 3P-STREC (thick lines) as well as their "95% CI + data uncertainty" (thin lines). **d** Comparison of D-STREC with observed MJJA temperatures (CRU dataset and southern Quebec historical record). Pearson's $r$-values in brackets are for 10-year low-pass filtered series (smoothed lines) using a Butterworth filter ($P < 0.01$ accounting for autocorrelation in time series).

target (Climatic Research Unit gridded Time Series version 4.03[17], hereafter CRU or CRU TS 4.03 dataset) reaches 0.75 (Pearson's $r$, $P < 0.001$) over the 1905–2006 calibration period and remains stable in two subintervals (Fig. 2d). A two-century historical record in southern Quebec (Methods) extends the period of verification, correlating at 0.40 and 0.62 ($P < 0.001$; 1837–2010 CE) with D-STREC at interannual and decadal time scales, respectively (Fig. 2d), even if this record was

developed ~450–1400 km away from the sites of our network (Fig. 1a). D-STREC also shares a high fraction of decadal variability ($r = 0.50$ over the 997–2006 common period, $P < 0.001$; Fig. 2c) with a previous summer temperature reconstruction of NENA based on independent proxy types (tree-ring width[18] plus $\delta^{13}C$ and $\delta^{18}O$ from tree-ring cellulose[19,20]; Methods), highlighting a well-reconstructed low-frequency domain over the last millennium. The spatial domain of

D-STREC (significantly positive $r$ with 10-year high-pass CRU temperatures, $P < 0.05$) covers the northeastern half of North America and has a striking resemblance with the correlation field of the regional MJJA temperature target, even outside the North American continent (Fig. 1b, c).

Compared to both 3P-STREC and its tree-ring-width component (Methods), the D-STREC reconstruction skill is substantially improved (Supplementary Table 3), with a much larger spatial domain (Fig. 1) and better-constrained uncertainties (Fig. 2c), due to the enhanced sensitivity of MXD data to high-frequency temperature variability[5,11,21] ($r_{\text{D-STREC}} = 0.72$ vs. $r_{\text{3P-STREC}} = 0.40$ with the 10-year high-pass temperature target). Furthermore, D-STREC agrees nicely with the historical record at 1816 CE following the 1815 CE Tambora eruption, although the recovery is one year longer for sites farther north (Supplementary Fig. 3). We speculate that this phenomenon reflects the increasing severity of post-eruption growth stress towards the northern treeline. In contrast, 3P-STREC underestimates this cooling by as much as -2.5 °C (Supplementary Fig. 3a), indicating that volcanic cooling is more reliably recorded by our MXD records. This evidence, in addition to the superior reconstruction skill, implies that D-STREC more precisely reconstructs cold conditions than 3P-STREC, for example, during -1600–1820 CE where the two reconstructions tend to differ (Fig. 2c).

## Millennial summer temperature history

D-STREC shows that the strongest centennial warming occurred during the twentieth and twenty-first centuries (1917–2016 CE), with a linear temperature rise of 0.16 ± 0.024 °C per decade (Supplementary Table 4). Furthermore, 2005–2014 CE was the warmest decade of the past 1246 years (1.25 °C above the 1905–2006 average; Supplementary Table 4). Warm summers also occurred during -950–1250 CE, a period coinciding with the Medieval Climate Anomaly present in temperature reconstructions and simulations at global and hemispheric scales[22]. With six out of the 10 warmest decades clustered between 950 and 1250 CE (Supplementary Table 4), medieval summers were, on average, 0.21 °C warmer than the 1905–2006 period. Conversely, relatively cold conditions prevailed from 1450 to 1930 CE (−0.52 °C with respect to 1905–2006 CE), a period that largely overlaps the Little Ice Age (-1450–1850 CE)[22]. The 1590–1930 interval represents an exceptionally cold phase in NENA (−0.69 °C), with nine out of the 10 coldest decades and more than half of the 44 coldest summers [≤mean−2 times the standard deviation (SD)] since 772 CE (Supplementary Tables 5, 6).

## Impact of volcanic eruptions

Cold decades during the past 1246 years were frequently related to large tropical eruptions. The 1815 CE Tambora eruption resulted in the second coldest summer (1816 CE, −4.53 °C with respect to 1905–2006 CE), and the coldest decade (1816–1825 CE, −1.70 °C), although this cold period may have been influenced by the 1822 CE Galunggung eruption. Other large tropical eruptions, such as the Huaynaputina (1600 CE), Parker (1640 CE), and Cosiguina (1835 CE) events, were each followed by very cold decades (ranked the 2nd, 4th, and 9th coldest, respectively), although additive effects of multiple events are also possible (Supplementary Table 5). Although the 1453–1462 CE interval is only ranked as the 22nd coldest decade, it is nevertheless the coldest decade between 1000 and 1600 CE, and corresponds to two large tropical eruptions during the 1450 s (1452 and 1457 CE). Conversely, the 1257 CE eruption of Samalas, the most sulfur-rich eruption of the Common Era[23] and associated with a pronounced summer cooling in Europe[24,25], only produced a weak cooling anomaly of −0.76 °C at 1258 CE (with respect to 1905–2006 CE) and a mean temperature of −0.23 °C during 1258–1267 CE. This moderate response is in line with the 3P-STREC reconstruction (Fig. 2c) and is probably due to region-specific volcanic responses[25] and complex atmosphere chemistry[26–28]. At the same time, the Samalas event belongs to a sequence of four

closely spaced strong tropical eruptions in 1229, 1257, 1275, and 1285 CE (Supplementary Table 7), which may have induced a long-term cooling trend in NENA (−0.06 °C per decade during 1200–1300 CE according to D-STREC), at the end of the Medieval Climate Anomaly[18,29].

In fact, at the annual time scale, out of the 44 coldest summers reconstructed by D-STREC, 15 and 16 years can be attributed to tropical and Northern Hemisphere extratropical (NHET) eruption events, respectively (Fig. 2c; Supplementary Table 6; Methods). The binomial-distribution test indicates a more robust attribution result since 1000 CE than 772 CE ($P = 0.021$ and 0.051, respectively; Methods). The higher proportion of unattributed cold extremes in the first millennium (5 out of 9) is most likely due to greater uncertainties in ice-core dating of earlier volcanic events[30]. Accordingly, we constrain our subsequent volcano-related analyses to the 1000–2017 period.

Superposed epoch analysis (SEA; Methods) confirms that tropical eruptions induced longer cooling episodes than did NHET eruptions. On average, tropical eruptions caused about a 5–10 years of significant cooling (0.95 significance level) followed by an additional -2-year recovery to the pre-eruption level (Fig. 3a). The cooling peak generally lagged the tropical eruptions by 1 year and corresponded to the forcing peak (Fig. 3; Supplementary Fig. 4a). In contrast, the cooling effect of NHET eruptions lasted only -1–3 years and was most frequently significant at the year of the eruption (Fig. 3b; Supplementary Fig. 4b). Comparisons of SEAs using multiple subsets of volcanic events validate the more persistent cooling following tropical eruptions (Fig. 3a, b; Supplementary Fig. 5), a result not affected by closely spaced eruptions. Analyses on NHET summer temperature reconstructions and simulations (Supplementary Fig. 6) further confirm the consistency of these results across the Northern Hemisphere. Stratospheric aerosols injected by tropical volcanoes spread poleward with a residence time of 1–3 years[31,32], while aerosols of NHET eruptions are mainly constrained to 30–90°N with a shorter lifetime[33,34]. Because tropical eruptions influence a larger oceanic domain with high thermal capacity[32,35], ocean–atmosphere heat exchanges can cool continental summers[36] beyond the direct aerosol forcing, in a more persistent way compared to NHET eruptions (Fig. 3). In contrast to D-STREC, 3P-STREC shows attenuated cooling peaks that lag tropical and NHET eruptions by about 9 and 5 years, respectively (Supplementary Fig. 7). This behavior most likely results from the strong biological memory of ring-width data[11], the only high-frequency component of 3P-STREC.

## Region-hemisphere coherence

Although D-STREC was developed from a limited sector of NENA, it behaves like a large-scale, NHET temperature reconstruction at the multidecadal time scale. The 20-year smoothed (Butterworth filter) D-STREC reconstruction correlates significantly with six NHET tree-ring-based summer temperature reconstructions (Fig. 4a; $r_{851–2000 \text{ CE}} = 0.39–0.56$, $P < 0.001$), of which five are mostly dominated by Eurasian tree-ring data[5,37–39] and one consists of hemispheric data after excluding the NENA domain[14] (Supplementary Fig. 8). Furthermore, warm and cold epochs of D-STREC correspond well with an ensemble of 25 full-forcing simulations of Northern Hemisphere MJJA temperatures developed following the Coupled Model Intercomparison Project Phase 5[40]/Paleoclimate Model Intercomparison Project Phase 3 protocol[41] (hereafter referred to as CMIP5, see Methods; Fig. 4b). The significant correlation of D-STREC with the 20-year smoothed CMIP5 multimodel mean ($r_{851–2000 \text{ CE}} = 0.55$, $P < 0.001$) exceeds those between D-STREC and the correlations with four NHET reconstructions. It even surpasses the correlations between two NHET reconstructions and the multimodel mean (Fig. 4c). The region-hemisphere coherence is particularly strong during 1000–1850 CE, and remains relatively high and significant at 20–100-year time scale (band-pass filter) even if centennial trends are removed (Supplementary Table 8; Fig. 4c), indicating a high fraction of synchronous multidecadal variability between NENA and hemispheric

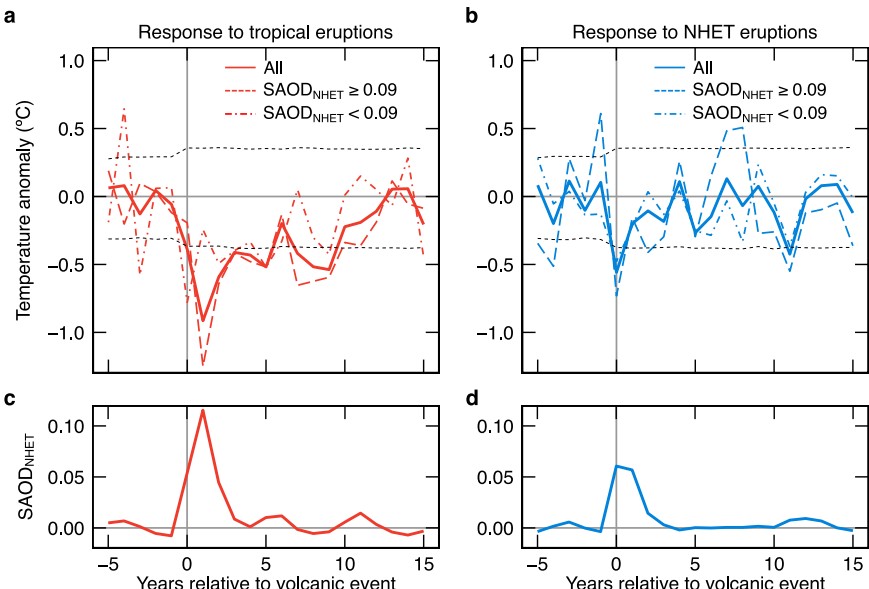

**Fig. 3 | Superposed epoch analysis for D-STREC according to volcanic aerosol forcing since 1000 CE. a** D-STREC responses to tropical eruptions with stratospheric aerosol optical depth at 550 nm area-weighted averaged over 30–90°N (SAOD$_{NHET}$) ≥ 0.03 ("All"), SAOD$_{NHET}$ ≥ 0.09, and 0.03 ≤ SAOD$_{NHET}$ < 0.09. **b** Same as **a** but for Northern Hemisphere extratropical (NHET) eruptions. **c** Superposed volcanic aerosol forcing (expressed as SAOD$_{NHET}$) of tropical eruptions with SAOD$_{NHET}$ ≥ 0.03. **d** Same as **c**, but for NHET eruptions. Temperatures and forcing are calculated relative to the 5-year pre-eruption mean. Horizontal dashed lines in **a** and **b** represent the 0.95 statistical significance level for the "All" group (Methods). The eruptions used for analyses are listed in Supplementary Table 7.

summer temperatures. The most prominent difference between D-STREC and simulations during the last millennium concerns the impact of the Samalas eruption (Fig. 4b). D-STREC provides complementary proxy evidence that the short-term cooling effect of the Samalas was disproportionately low compared to its amplitude in CMIP5 climate model simulations[8,28,37].

By averaging 25 members, the CMIP5 multimodel mean largely masks out the unforced internal variability[7,42–44]. Thus, the significant correlations among D-STREC and NHET temperature reconstructions and the multimodel mean simulation imply that externally forced climate variations produce a strong imprint at both regional and hemispheric scales. This is also well supported by strong NENA-NHET coherence in individual CMIP5 simulations (Supplementary Table 8). Furthermore, D-STREC shows the highest correlations with the volcano-only ensemble among single-forcing simulations of NHET land summer temperatures (Supplementary Fig. 9). This result, along with SEA (Fig. 3; Supplementary Fig. 5) and correlation analysis after excluding post-eruption years (Fig. 4d, e), points toward volcanism, in particular tropical eruptions, as the main forcing synchronizing NENA and hemispheric summer temperatures. Consequently, in conformity with detection and attribution studies on hemispheric and global millennial temperatures[2,45,46], our study based on high-quality proxy data highlights the dominant role of tropical volcanism in shaping multidecadal temperature variations across spatial scales. Predictability of multidecadal temperature variability thus remains challenging without any information on future volcanism.

## Methods
### MXD network and chronology development
Our MXD network consists of series from 1668 radii of 1294 black spruce [*Picea mariana* (Mill.) B.S.P.] trees from four sites across the eastern Canadian boreal forest (Fig. 1a; Supplementary Table 2), an extratropical region with typically cold/long winters and warm/short summers. L105, L20, and L135 are three newly sampled sites. To ensure data homogeneity[18,47], we sampled living trees from the lakeshore forests of corresponding lakes where subfossils were collected. New MXD data from these sites were measured from 1–2 radii of each sample using the X-ray densitometric technique (see Wang et al.[48] for

details). The dating of millennial chronologies at these three sites was validated using a subfossil wood sample showing a globally coherent cosmogenic ¹⁴C signature at 774 CE[49]. The existing Quex dataset was obtained from the International Tree Ring Data Bank (National Oceanic and Atmospheric Administration, NOAA; Supplementary Table 2) and was corrected for its erroneous location in the metadata and for the cross-dating of one sample (Supplementary Method 1). All MXD measurements were averaged by the tree before subsequent analysis.

After comparing three standardization methods, including the widely applied regional curve standardization[50], the regionally constrained individual signal-free standardization (RSFi)[51] method was selected to detrend the MXD series at each site (Supplementary Methods 2, 3). The RSFi method efficiently removed nonclimatic signals (e.g., local competition and disturbances) introduced by trees established in different eras, while preserving the long-term variability. In addition, because the MXD series of black spruce trees are known to exhibit heteroscedastic variance[21], we compared chronologies calculated from ratios and residuals plus power-transformation[52] via a time-efficient linear scaling reconstruction approach[53]. The RSFi ratio chronologies were chosen for the final D-STREC temperature reconstruction due to better overall performance (Supplementary Method 2). Chronology characteristics were assessed by the EPS[16], rbar, and mean cambial age (Supplementary Fig. 2).

### MXD-based MJJA temperature reconstruction
The D-STREC summer temperature reconstruction was developed using a Bayesian linear regression approach[19] (Supplementary Method 4). Compared to conventional reconstruction methods, the Bayesian approach provides posterior distributions of the climate variable, taking into account individual proxy likelihoods, thus enabling comprehensive uncertainty assessments and improving the reconstruction skill (Supplementary Fig. 10). The four MXD chronologies showed optimal temperature responses over the MJJA season of the current growing year (Supplementary Fig. 11). Therefore, regional MJJA temperatures were averaged over an area covering our data network (50–60°N, 65–77°W) from the CRU TS 4.03 dataset[17] (Fig. 1a), and the full calibration period was set to 1905–2006 CE, which is the time interval common to the three longest MXD series plus the

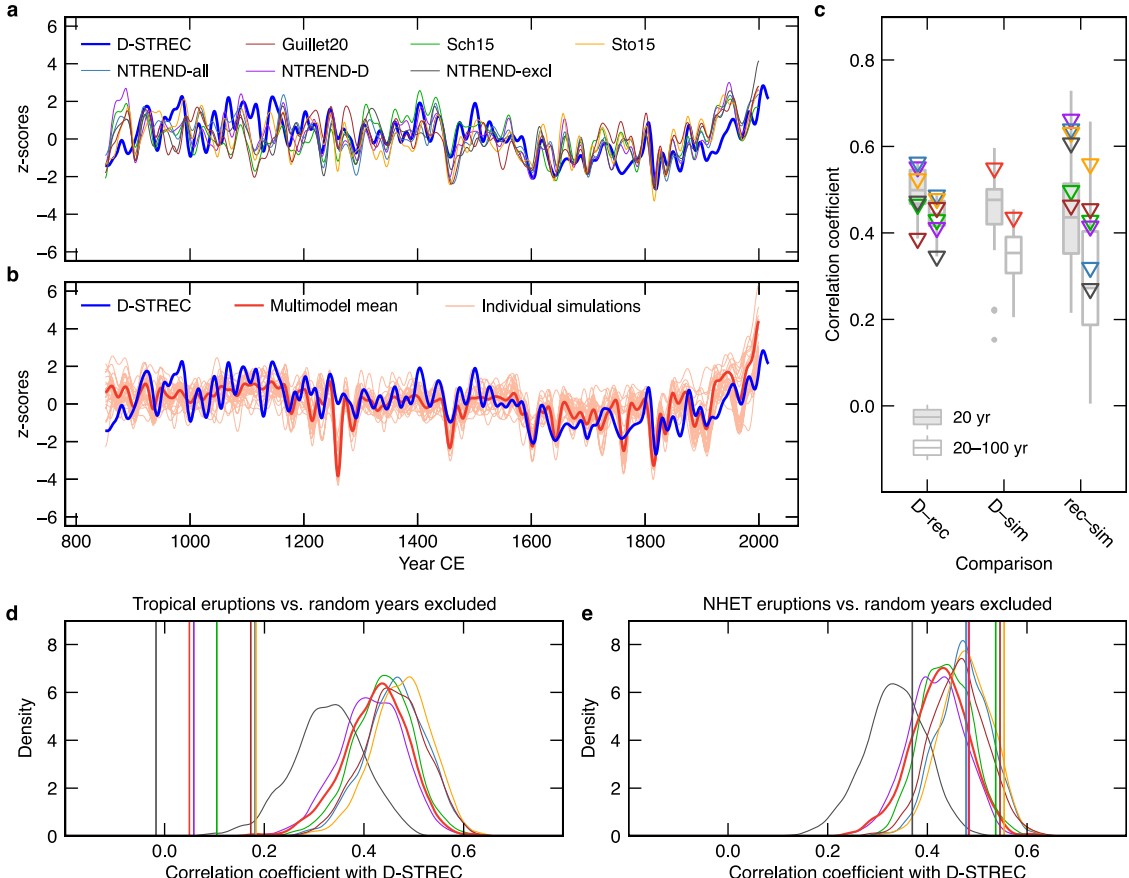

**Fig. 4 | Comparison of D-STREC with Northern Hemisphere extratropic (NHET) summer temperature reconstructions and simulations. a** D-STREC compared to six tree-ring-based reconstructions. Guillet20: Guillet et al.[39]; NTREND-all and NTREND-D: the all-proxy and the maximum latewood density only reconstructions in Wilson et al.[5], respectively; NTREND-excl: Northern Hemispheric mean of the gridded NTREND reconstruction[14] after excluding the northeastern North American components (Supplementary Fig. 8); Sch15: Schneider et al.[38]; Sto15: Stoffel et al.[37]. **b** D-STREC compared to an ensemble of 25 CMIP5 full-forcing near-surface May–August temperature simulations averaged over land 30–90°N. In **a** and **b**, all the time series are smoothed using a 20-year low-pass Butterworth filter to highlight multidecadal and long-term variations, and are transformed to z-scores with respect to the 1000–2000 time period. **c** Pearson's *r* among the D-STREC and NHET summer temperature series. 20 yr: 20-year smoothed series over the 851–2000 common period; 20–100 yr: 20–100-year band-pass filtered series over the 1000–1850 period. D-rec: D-STREC versus NHET reconstructions (boxes and triangles); D-sim: D-STREC versus individual simulations (boxes) and the multimodel mean (red triangles); rec-sim: NHET reconstructions versus individual simulations (boxes) and the multimodel mean (triangles; colors refer to the legend of **a**). Boxes show the median and the 25–75% range, while whiskers and points refer to the 1.5 times interquartile range and outliers, respectively. **d** Pearson's *r* between D-STREC and NHET multidecadal temperature reconstructions and the multimodel mean after excluding 10 years following major tropical eruptions (vertical lines) versus randomly selected years (density curves) (Methods). Correlations are computed using 20–100-year band-pass filtered series (1000–1850 CE) to avoid effects of data quality in the earlier period and recent anthropogenic warming. **e** Same as **d**, but for major NHET eruptions. Colors refer to the legends of **a** and **b**.

independent 3P-STREC reconstruction (used on a comparison basis, see below). Calibration of the fourth MXD chronology at the site Quex was limited to the 1905–1989 period, due to the shorter time coverage. D-STREC was limited to the 772–2017 period to ensure reconstruction robustness, based on EPS > 0.85 and replication ≥15 trees (Supplementary Fig. 2a). The final D-STREC reconstruction was derived from the median posterior density of each year. The correlation between the final D-STREC and MJJA temperature target is 0.76 and 0.75 for the 1901–2017 and the 1905–2016 periods, respectively.

Two types of confidence intervals were calculated for the D-STREC reconstruction. The first type (referred to as "95% CI") assesses the uncertainties of the Bayesian model and was derived directly from the 2.5th and 97.5th percentiles of the posterior temperatures for each reconstructed year. The second type (referred to as "95% CI + data uncertainty") additionally considers the time-varying uncertainties in the proxy chronologies. We produced 100 chronologies per site with a sampling procedure based on available MXD data points. Assuming that data points of individual trees are normally distributed for each year, the chronologies were built by sampling each year from a normal distribution N ~ (μ, σ), where μ is the mean of the

available data points and σ is the standard error of mean. Supplementary Fig. 1 illustrates the range of ±1.96 × standard error of the mean for the four local MXD chronologies. The 100 chronologies were then included in the Bayesian models for generating alternative D-STREC reconstructions. The final "95% CI + data uncertainty" was derived from the 2.5th and 97.5th percentiles of the mixed posteriors from the 100 runs.

**Comparison with the 3P-STREC temperature reconstruction**
We compared D-STREC with the earlier 3P-STREC reconstruction developed from published millennial ring width[18], δ13C[19], and δ18O[20] data from the same region near site L20. 3P-STREC and D-STREC are completely independent by proxy types (no proxy in common) and almost independent by sampling sites (one out of 9 sites in common). 3P-STREC comprises one high-frequency component (from ring-width data) and three low-frequency components (ring width, δ18O, and δ13C data). To allow direct comparison, we recalibrated the 3P-STREC dataset following the same Bayesian procedure and same MJJA temperature target (which correlates significantly with the three individual proxies; Supplementary Fig. 11) as for D-STREC. Similarly, two types of

confidence intervals were produced for the 3P-STREC. The ring-width component of 3P-STREC (3P-STREC-TRW), which is a variant of the dataset originally used to develop the first tree-ring-based millennial summer temperature reconstruction in NENA[18], was additionally compared with D-STREC using a linear scaling approach[53] and the aforementioned regional temperature target.

## Historical temperature record

We generated a long MJJA temperature record from a compilation of historical daily temperature observations from the Saint-Lawrence Valley in southern Quebec[54]. The historical record was constructed based on multiple observers in Quebec City, Montreal, and the Ottawa River region, which are ~450–800 km away from our closest site, L105 (Fig. 1a). Available maximum and minimum temperature data were averaged to represent the daily mean temperatures, which were later aggregated to monthly data. In order to minimize uncertainties caused by a high frequency of missing values in the early 19th century, a monthly aggregate for each year was retained only if there were less than eight missing daily values in the corresponding month. The valid temperatures for 4 months from May to August were then averaged to yield a seasonal (MJJA) temperature record starting from 1805 CE, with continuous values since 1837 CE (Fig. 2d).

## Last-millennium temperature simulations

We used 25 full-forcing and 23 single-forcing last-millennium simulations of monthly near-surface air temperatures. The full-forcing CMIP5 simulations include 16 members from the CESM Last Millennium Ensemble[55] (CESM-LME; including 3 members of isotope-enabled CESM[56]), and 9 members from the CMIP5 Past1000[41] and corresponding historical[40] experiments. The single-forcing simulations were obtained from the CESM-LME runs singly forced by greenhouse gases, land use, orbital, solar, and volcanic forcing. In addition, the preindustrial 850 control run of the CESM-LME was used as an unforced baseline to evaluate correlations between D-STREC and the single-forcing simulations. Corresponding climate models and experiments are detailed in Supplementary Table 9. In order to allow for direct comparisons among models, all model outputs were interpolated to a T21 resolution (~5.6° × 5.6°) using the first-order conservative remapping function provided by the Climate Data Operators[57]. Gridded data were then averaged over land between 30°N and 90°N according to area weights to generate a full-forcing and a single-forcing ensemble of simulated NHET land temperatures for the MJJA season. Although full-forcing CMIP5 simulations of Northern Hemisphere summer temperatures tend to underestimate the impact of NHET eruptions (Supplementary Fig. 6), this result probably reflects the fact that CMIP5 volcanic forcing differs from the updated eVolv2k and CMIP6 datasets we used to select volcanic events (Supplementary Tables 7, 9).

## Correlation analysis and significance test

We used Pearson's correlation to assess relations among reconstruction, simulation, and climate time series. Because strong autocorrelation reduces effective degrees of freedom of time series and could bias conventional Student's t-tests[58], we adopted the method of PAGES 2k[2] to test statistical significance for Pearson's r, with a null hypothesis that original time series are unrelated. First, we generated 1000 random red-noise time series for each original series with the same lag-1 autocorrelation coefficient using the "colorednoise" R package[59]. The random series were smoothed, if needed, and then correlated against each other as we did for the original data to form a distribution of $5 \times 10^5$ correlation coefficients for each pair of comparisons. Finally, these distributions were compared with the true Pearson's r to calculate probabilities (P-values) to test the null hypothesis under a two-sided test. Correlation coefficients are considered significant when P is smaller than 0.05.

## Attribution of cold extremes to volcanic eruptions

The cold extremes of D-STREC (≤mean−2 SD) were attributed to volcanic eruptions according to three ice-core-based volcanic forcing reconstructions (eVolv2k[30], IVI2[60], and ICI[61]). Locations of eruptions were identified from the corresponding volcanoes provided by the confirmed eruption list (dating uncertainty ≤1 year) of the Global Volcanism Program[62] (GVP; tropical: 30°S–30°N, NHET: 30–90°N) or from the ice-core datasets (for unidentified events). Eruptions from Southern Hemisphere extratropics were not considered since they have negligible climatic impacts on the extratropical Northern Hemisphere[63]. Similar to SEA, we updated dates for unidentified eVolv2k events according to Toohey et al.[64], which are more likely related to true eruption dates (see below). We retained a total of 298 events by removing duplicated eruptions, including those matched by Toohey et al.[30], from the three eruption datasets. A successful match was identified when a cold extreme corresponded to a retained volcanic event, allowing a maximum 2-year lag prior to the cold year, accounting for time lags of volcanic responses and uncertainties in ice-core dating. If several closely spaced eruptions were matched to an extremely cold year, the eruption of the largest magnitude was considered. The attribution result based on the original, date-unadjusted eVovl2k plus IVI2 and ICI datasets is identical to that using the adjusted eVovl2k (Supplementary Table 6). The probability that the observed match frequency differs from a random result was estimated from the binomial distribution. Because of the greater uncertainties in ice-core dating of earlier eruptions[30], we tested the statistical significance of the attributions for the entire reconstruction and for the period after 1000 CE.

## Superposed epoch analysis (SEA)

We used a regular SEA approach provided by the algorithm of Rao et al.[65] to investigate the composite responses to the target volcanic events. First, we constructed a list of tropical and NHET volcanic eruptions (Supplementary Table 7) according to the stratospheric aerosol optical depth at 550 nm (SAOD) estimated from the eVolv2k reconstruction (1000–1900 CE)[30] combined with the Coupled Model Intercomparison Project Phase 6 (CMIP6) volcanic forcing dataset (1901–2016 CE)[66]. Events in the first millennium were not considered for SEA due to greater dating uncertainties[30]. The SAOD$_{NHET}$ (area-weighted average over extratropical 30°N to 90°N) ≥ 0.03 (~1/3 of the Pinatubo 1991 CE eruption) was used as criteria by which to select tropical and NHET eruptions that have potentially affected the extratropical Northern Hemisphere. We kept all the events that corresponded to identified eruptions with volcanic explosivity index (VEI) ≥ 4 according to the GVP[62]. The unidentified events were further screened, and were included when also listed in both IVI2[60] and ICI[61], by permitting a 3-year lag[67]. Four events that fulfilled the criteria (Aira 1471 CE, Serua 1693 CE, Unidentified 1808 CE, and Pinatubo 1991 CE) were discarded because of case-specific reasons (Supplementary Table 7). In total, 24 tropical and 19 NHET events were retained for the SEA.

The key eruption years used for the SEA were re-evaluated to minimize potential uncertainties in the assessment of volcanic effect. For unidentified events, we adopted the years adjusted by the latest application of eVolv2k[64], accounting for the time lags between eruptions and ice sheet deposition. The adjustments on 10 tropical events led to a more consistent SEA result compared to that using identified tropical eruptions (Supplementary Figs. 5a, 12a). In addition, key years were set 1 year after the eruption for events that occurred in or after August (otherwise assumed in the same year), a strategy adapted from Guevara-Murua et al.[68]. Not considering this lag could introduce biases to SEA because tree ring is a seasonal proxy. For example, an eruption in December cannot affect the tree-ring formation in the same year because black spruce grows in spring–summer[69]. This adjustment was applied to limited events with known eruption months, yet, it could result in a more evident cooling in response to NHET eruptions

(Supplementary Fig. 12b). We performed SEA on multiple time series (reconstructions and simulations) based on the above-constructed key eruption years. We considered the temperature anomalies of 15 post-eruption years relative to the 5-year pre-eruption mean. The statistical significance of volcanic cooling was assessed using the block reshuffling method[70] with 10,000 iterations.

### Region-hemisphere coherence vs. tropical and NHET eruptions

We designed an additional experiment to compare the impact of tropical and NHET eruptions on the coherence of summer temperatures between NENA and the Northern Hemisphere. We correlated D-STREC with seven NHET temperature series (six reconstructions and the CMIP5 multimodel mean; Fig. 4a, b) after excluding 10 years (the year of eruption included) following each tropical or NHET eruption. This analysis considered the SEA eruption list (key years in Supplementary Table 7) but excluded the period earlier than 1000 CE and the recent warming trend after 1850 CE. In total, 173 and 143 post-eruption years were excluded from each series to form a nontropical and a non-NHET eruption group, respectively. The retained values in individual temperature series were then chronologically stitched, smoothed using a band-pass filter (20–100 years), and correlated with the similarly processed D-STREC. Although stitching time series is somewhat arbitrary, the resulting correlations help assess the impact of eruptions on the region-hemisphere coherence. We produced 1000 sets of 19 pseudo-tropical and 15 pseudo-NHET eruptions by randomly sampling without replacement between 1000 and 1841 CE, of which the number of post-eruption years to be excluded was the same as corresponding true eruptions. By excluding 10 years following each set of pseudo-eruptions, we obtained 1000 correlation coefficients between D-STREC and each NHET series.

### Data availability

The final D-STREC reconstruction generated in this study is available at https://www.ncei.noaa.gov/access/paleo-search/study/36574. Tree-ring maximum latewood density data of sites L20, L105, and L135 are available on the NOAA World Data Center for Paleoclimatology (https://www.ncei.noaa.gov/products/paleoclimatology; studies 36575, 36576, and 36577). Northern Hemisphere tree-ring reconstructions (Anchukaitis et al.[14], Guillet et al.[39], Schneider et al.[38], and Wilson et al.[5]) are available at https://www.ncei.noaa.gov/products/paleoclimatology. The Stoffel et al. reconstruction[37] was derived from https://www.blogs.uni-mainz.de/fb09climatology/files/2018/06/NH-reconstructions.xlsx. CRU TS 4.03 data were obtained from KNMI Climate Explorer (https://climexp.knmi.nl/). Quebec historical temperature observations are available at https://www.ncei.noaa.gov/access/paleo-search/study/16336. Data URLs for other datasets used in this study can be found in Supplementary Information. Wood samples are stored in the Laboratory of historical ecology and dendrochronology at Université du Québec à Rimouski (contact: D.A., dominique_arseneault@uqar.ca).

### Code availability

The code used to analyze data is available from the corresponding author upon request. The code for superposed epoch analysis is available in Rao et al.[65].

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

## Acknowledgements

F. Vignola, A. Delwaide, N. Trou-Kechout, P. Balducci, and J. Larose contributed to collecting and processing black spruce samples.

A. Schurer and B. Nasri kindly commented on the earlier version of this article. R. Liu helped with the tree-ring density measurements. S. Payette and J. Esper provided the Boniface River and Greece tree-ring data, respectively. M. Toohey provided the combined eVolv2k and CMIP6 volcanic forcing dataset. M.P. Rao provided and interpreted the SEA algorithm. This work was supported by the Natural Sciences and Engineering Research Council of Canada in collaboration with Hydro-Québec, Manitoba Hydro, and Ouranos under the PERSISTENCE project (grant RDC 485475-15 to É.B. and D.A.). F.W. also received support from the China Scholarship Council. F.G. was supported by the Natural Sciences and Engineering Research Council of Canada (grant RGPIN-2021-03553). S.Y. and T.Z. were supported by Science and Technology Department of the Xinjiang Autonomous Region (grants 2020D04040, 2018D04028).

## Author contributions

F.W., D.A., and É.B. conceptualized the study. F.W., D.A., É.B., and F.G. conducted the field work, performed the analyses, and wrote the manuscript. F.W., S.Y., and T.Z. measured tree-ring density data. All authors have reviewed and helped to revise the manuscript.

## Competing interests

The authors declare no competing interests.
