## [Peer Review File · Nature Communications]

Tropical volcanoes synchronize eastern Canada with Northern Hemisphere millennial temperature variabilityREVIEWER COMMENTS

Reviewer #1 (Remarks to the Author):

Overview

This study by Wang et al. develops new MXD chronologies and reconstructs the last millennium summer temperature of the Northeastern North America (NENA) region. Their new reconstruction shows stronger and less delayed volcanic responses than an existing reconstruction based on ring width and cellulose, confirming the superiority of MXD as a recorder of high frequency temperature variation. Then, they combine these two reconstructions to form a new one for NENA that shows overall higher skill, and find a good agreement with climate model simulated Northern Hemisphere extratropical (NHET) summer temperature regarding temporal variation. The comparison to single-forcing simulations indicates that tropical volcanic eruptions are the main forcing that synchronizes the NENA and hemispheric summer temperatures.

In my opinion, these results are of importance and interest to the community, and can stimulate further studies. The manuscript is overall in good quality, with a clear structure, and analyses being thorough and to the point. I have only a few minor comments that I list below. Once those have been addressed, I recommend the work be accepted for publication.

Details

Regarding Figure 4b:

+ This figure shows the comparison of the reconstruction 4P-STREC and the model simulated summer temperature over land 30-90N, using a 20-yr low-pass filter. What is the purpose of applying a low-pass filter here, and why 20-yr? Explanations are needed in the main text.

+ Out of curiosity, what is the comparison like between the reconstruction 4P-STREC and the model simulated NENA summer temperature? A better consistency is expected, but it could still be a useful apples-to-apples comparison to be displayed in the supplementary information.

+ What is the comparison like between the model simulated NENA and NHET summer temperatures? If the NENA-NHET synchronization exists in model simulations, it could also support the main message of this study.

Reviewer #2 (Remarks to the Author):

This paper presents a new tree-ring maximum latewood density (MXD) series from northeastern North America covering the last millennium. Such data is very, very welcome and quite exciting, as this region is still a data-gap in terms of high signal-to-noise proxy data for Common Era temperature reconstructions (as the authors accurately point out). These data will be extremely valuable for enhancing large-scale temperature reconstructions. The analysis here is interesting in several ways, particularly in how it extends, modifies meaningfully, and enhances previous analyses using non-MXD records from the same trees, species, and region. All the methods seems reasonable to me - mostly they are standard dendrochronological methods and where necessary certain innovations are used to deal with the challenges of this particular species and collection. But these all seem reasonable. I do always have some concerns about the use of Signal Free detrending, in that it can definitely cause (ironically) end-effect biases under certain circumstances and for certain datasets. However, I do not see any obvious evidence for the existence of these biases in this case.

My major concerns are about the strength of claims about the insignificance of internal variability and how this is supported (or not) by the data and analyses shown in the manuscript. I don't see any reason to think these concerns would be ultimately disqualifying for the manuscript, but I would like the authors to consider my substantive and minor comments below and to reconsider the strength of some of their claims or statements about internal vs forced variability.

1. I'm curious about the motivation of combining the new MXD data presented here with the weaker temperature proxies in 3P-STREC (e.g. Line 98 and 99). What utility is there in adding less sensitive proxy data to a much improved MXD temperature reconstruction, especially given the complex mechanistic relationships between isotopes and temperature? The authors (Line 93 and Line 101) claim the the four proxy reconstruction is better based on correlation values, but the difference is only $r=0.72$ vs. $r=0.77$, and it is unlikely that the magnitude of difference between these two values is actually significant (which can be measured with a Z test), particularly since the gain comes while adding additional series (which would inflate the variance explained simply because of the additional predictors). Why not simply use the D-STREC reconstruction (and then continue as in various places in the supplemental material to compare/contrast with the earlier 3P-STREC and STREC reconstructions from Gennaretti?). In particular, it would be useful to compare the similarities and highlight the differences between the tree-ring STREC reconstruction, the multiproxy/isotope 3P-STREC reconstruction, and the current D-STREC MXD reconstruction independently because each of these has been previously put forward as a regional

temperature reconstruction and earlier conclusions about e.g. volcanic eruptions and climate cooling have been drawn from these as well. How do these three generations of Quebec black spruce reconstructions differ, particularly with respect to the impact of volcanic eruptions like Samalas etc., once outside of the calibration period? For instance (Line 125-126), the current reconstruction has Samalas cooling as relatively small, but the original STREC reconstruction from Gennaretti et al. highlighted cooling around the time of Samalas as one of the major results of that earlier paper a la Miller et al. Similarly, in Lines 132-134 the authors find an extended significant cooling following tropical eruptions using SEA, which is similar to mixed proxy reconstruction superposed epoch analysis but different from MXD-only reconstructions like that of Schneider et al. 2015. In Figure S7, the separation of D-STREC from 3P-STREC suggests the latter is biased in its recovery of the sharp cooling related to volcanic events.

2. On volcanic SEA: Figure 3: The claim in the manuscript is that the cooling post-tropical eruption last for a decade, but in Figure 3 the SEA composite crossed back into 'non significant' after just a few years. In Figure S6 and S12, SEA typically involves a resampling (random with replacement or block) to assess significance levels, but that doesn't seem to have been done here as it was in Figure 3. Would the use of conventional SEA confidence intervals change the inference about the length of cooling in the supplemental figures and in the manuscript? This question is even more important since Figure 3 and Figure S7 does have these types of confidence limits and both suggest a return to within the baseline of cooling by 3 to 5 years other thereabouts. Figure S3 also shows that for the very large eruption of Tambora the reconstruction returns to baseline within 3 years. I'm concerned that something about the SEA (probably the event list and/or the significance testing) seems to suggest a longer recovery from a single event than is warranted. This not to say that multiple events (e.g. in the 1450s) couldn't combine to cause a ~decade of cooling, but rather that the recovery from a single event needs to be looked at more closely, particularly if the single large eruptive event in 1815 shows a rapid recovery (including in the historical data shown in Figure S3). And in Figure 3, Figures S11, S12, and Table S7: The event lists here contain several closely spaced eruptions, which could bias the SEA since there are actually multiple events in the post event window - how would the SEA (e.g. in Figure 3) change if only the very largest events (e.g. Tambora or Pinatubo or larger) with at least 10 years of spacing were considered?

3. Excessively strong claims about the importance of forced vs. unforced (decadal) variability. The authors based their claims that there is little unforced decadal variability at regional scales on Figure 4. Central to this claim is the correlations between low pass 20-year smoothed Northern Hemisphere reconstructions. The authors claim this provides evidence that at decadal scales their regional reconstructions matches large-scale reconstructions. However, there are reasons to approach this with caution: (1) as noted below, smoothing the series with a low-pass filter will enhance the correlation but reduce the degrees of freedom and this requires a modification of the significance level (or, more likely, some sort of simulation a la Ebisuzaki 1997) to asses a more accurate null; (2) Nearly all of the reconstructions considered us the STREC tree-ring reconstruction (NTREND, Stoffel and Guillet) - so the regional reconstruction includes information that is in the large-scale reconstructions. The lowest correlation between 4P-STREC and an existing reconstruction is somewhat worryingly with the Schneider et al. 2015 reconstruction, which is the only one to use pure MXD besides NTREND_D.

Moreover, reconstructions including NTREND use an MXD only chronology from Quebec back to the 14th century. So, there is some degree of overlap in the fact that the large-scale reconstructions contain regional information that is likely reflected in the 4P-STREC reconstruction; (3) It is clear that while there is strong synchronicity between the regional reconstruction and the large-scale reconstruction during certain periods associated with multiple volcanic eruptions and widespread cooling (1450s, 1600, early 1800) there is disagreement at other decadal periods, particularly prior to 1300 CE (as highlighted by e.g. Esper et al. 2018 in *Dendrochronologia*). So, there is indeed support here (and elsewhere) for the idea that decadal scale cooling is synchronous for several periods of volcanic activity (again, particularly for periods during the LIA, partial coincident with solar minima, and specifically around 1450s, 1600, and early 1800s), but it is simply too strong to use the evidence here to downplay the role for internal climate system variability in influencing the magnitude of cooling and spatial patterns of cooling. I would be happy for the authors therefore to suggest that volcanic eruptions synchronize decadal scale cooling, but the paper does not support the idea that internal climate system isn't important across a range of timescales and even during cold periods. In fact, one can see that (accounting for poor data coverage in many locations) in the Neukom et al. paper on non/synchronous warm and cold periods that what is different is not so much that places aren't cooler during periods of explosive volcanism, but rather the degree of cooling is different - with some places showing e.g. the 1600 or 1450s as coldest, and other regions showing different periods a relatively the coldest (this is a problem with the Neukom analysis, in addition to issues of data quality in those spatial reconstructions). Taking a closer look at NTREND, where data quality isn't as much of an issue (although data availability still is) one can see that the coldest years or decades are all associated with volcanism, but which period is the COLDEST varies in space (perhaps because of noise in the proxies, but also likely because of the influence of the combined nature of the forced plus unforced variability). Thus the claims in e.g. Lines 175 to 180 and the central framework of the paper (e.g. that internal climate system isn't important for decadal patterns of variability) are overstated.

Additional comments

Line 71: remove 'unprecedented'

Line 81, 87, Figure 1: Do these statistics account for the influence of trends in the datasets (which tends to inflate both the effect size as well as the regional area of significance)? How does the effect size (r) and significance level change if only interannual variability is considered (high pass filtered)? Based on the global pattern of correlation, I suspect the field correlations shown in Figure 1 include the large-scale warming trend, which will bias these field correlations toward the appearance of large-scale coherency (e.g. see Figures 4 and 5 in Anchukaitis et al. 2017 in *QSR*).

Figure 1: How is the inset map in Figure 1a created? Presumably not all of these sites were correlated against the field individually - so is this the average of the MXD sites shown? As above, have you accounted for autocorrelation and trend and its effects on these types of field correlations?

Line 101: '96% of the CRU observational target constrained within its 95% confidence interval' - but wouldn't you expect this by definition (e.g. that the confidence interval at a certain level contains the target data at that level)? I'm unclear why this is particularly diagnostic, at least as described.

Line 114, 115: '~1251–1930 CE' - this is an unusual definition of the LIA, particularly the idea of extending it to the 1900s! - why not use the IPCC definition of 1450 to 1850 CE? I see this reconstruction has anomalous cooling into the 1900s indeed, but that doesn't mean a quasi-global definition of the LIA shouldn't be applied.

Line 121: there is some doubt if this Kuwae - see e.g. Hartman et al. 2019 and the discussion and citations therein

Line 127-128: Perhaps, maybe, but there are lots of other uncertainties about Samalas that should be mentioned here related to the volcanic forcing - e.g. see for instance at least Timmreck et al. 2009 in GRL, Marshall et al. 2019 in JGR, Wade et al. 2020 recently in PNAS, and specifically with regarding ENSO and Guillet's hypothesis, the papers by Dee et al. in Science in 2020 and Zhu et al. in Nature Communications in the last few months (which do not support an El Niño event at that time)

Line 153: Does the significance level here account for the autocorrelation and loss of degrees of freedom induced by the smoothing? I think so, based on Lines 292-304, but I am somewhat surprised at this level of significance, given the loss of degrees of freedom

Line 167: omit 'relatively high' - could substitute with 'significant' (assuming as per my comment above accounting for the loss of degrees of freedom still leaves these relationships significant)

Line 175: But Mann et al. is specifically talking about the AMO, which is difficult to separate - and indeed projects onto - the large-scale Northern Hemisphere mean. This study, which is specifically about summer air temperature in a particular region compared to other large-scale reconstructions, says nothing in particular about the dynamics driving the AMO. This claim here can therefore be omitted, since it is not directly relevant.

Line 308: Why use VEI as a criteria, when VEI does not correspond well to climatic forcing (which is related to SAOD)?

Table S1: What would a similar analysis to Figure 4 look like with the individual MXD records in this Table? I suspect that most of these show the major cooling events around the 1450s, 1600, and early 1800s, since these appear across large-scale reconstructions and in spatial reconstructions like NTREND and Guillet, but this would provide a way to look at the ratio of common decadal-scale forcing between widely separated series not sharing any data or even regional similarities

Figure 4: What if instead of a low-pass filter you used a band-pass to isolate 20 to (for instance) 40 year periods (and remove both the high and low frequencies)? Does this give the same impression as the statistics here, or is centennial-scale variability (left in place by the low-pass filter) influencing the correlations at all here?

Reviewer #3 (Remarks to the Author):

Comments to Wang et al "Tropical volcanoes synchronize eastern Canada with Northern Hemisphere millennial temperature variability"

The manuscript presents an exceptionally well-replicated regional network of maximum latewood density data from eastern Canada. The region is somewhat underrepresented in hemispheric millennium long temperature reconstructions 1) because so far only ring width comfortably includes MCA and this proxy has documented weaknesses compared to MXD, 2) because MXD previously sampled in the region only reaches back to mid-1300 CE with modest replication, 3) because stable isotopic tree-ring data was never produced on an annual scale, which severely challenges any analyze of volcanic signatures. The new data is in itself a substantial contribution to the field of late Holocene paleoclimatology. The new temperature reconstruction which has substantial advantages over previous attempts (chronology length, signal strength, reduced confidence intervals due to the large replication and varied spatial representation and proxy type) is ideally suited to explore volcanic cooling and its overall effects on temperature variations.

The authors find that the variability of the reconstruction is largely driven by volcanism, at least prior to when GHG started to become a major forcing agent. Even comparable to the volcanically induced variability in NH products from either tree-rings or climate models. The amount of volcanic cooling that cuts through the stochastic nature of internally forced climate is remarkable. However, it is also not

completely surprising since the included sites from the region overall express a quite low inter-series correlation (r_{bar}) FigS2, which means that common features such as volcanic cooling will be high-graded, similarly to what is observed for NH tree-ring based reconstructions and climate models. The reconstruction thus serves as very valuable regional constraint of volcanic cooling forcing for climate models. I particularly think of Samalas 1257 which only modestly impact the temperature here. Here the models and the 4P-STREC differ substantially.

The manuscript is well written, carefully worked through with overall sound methodology. My main concern regards the inclusion of 3P-STREC. I understand the rationale because the T signal is strengthened in the lower frequencies by introducing more data from different sources, but the volcanic signature does not benefit from its inclusion and this is the main part of the manuscript. The authors try to tackle this problem by decomposing the signals in low and high frequency components but I worry that this will anyway smear the possibly more distinct signature. Moreover, the author could potentially extend the volcanic analysis to cover >200 yrs more when using only the D-STREC. I guess I am trying to say that it would be better if the T-reconstruction would be done using 4P-STREC, and the volcanic analysis in its full extent relied more heavily on only D-STREC.

It would also be more transparent to present temperature calibrations for the full D-STREC overlap with T-data, that is, up to 2017.

Below I list some minor comments:

L63-68 Perhaps refrain from revealing the conclusions in the introduction. The passage could be rephrased to “Based on this dataset, we test the proportion of forced and stochastic variability... on the regional reconstruction... and compare to that of NH hemispheric...”

L81 Why was the instrumental MJJA only presented up to 2006? Also in Fig S9. Is the replication of the reconstruction fading after this or is it something with the instrumental data?

L94 Should mention that the negative spike in t+1 is nicely reproduced by the reconstruction but t+2 appears to exhibit a memory from t+1, and that the recovery to track instrumental T only occurs in t+3. That is, there may be some memory effect (Esper et al., 2015) also in the D-STREC and 4P-STREC? Whereas the 3P-STREC clearly has a muted cooling but also a lagged recovery..

114 not necessary to present the abbreviation of an expression, remove

L114 This is a pretty extended definition of the LIA. Present what the general definition of the LIA is with a reference, and then you can go into the regional expression from your reconstruction.

Table S3 I note that although you split the calibration period into two periods, I cannot find any statistical tests for validating the calibration, and thus reconstruction. Please include statistics such as Reduction of error (RE) and coefficient of efficiency (CE) as a minimum, but also consider using the stationarity test discussed in Wilmking et al., 2020 GCB.

L121 See Esper et al., 2017 Bulletin of Volcanology, for a better date for Kuwae

L122-123 The Santa Maria eruption did not seem to produce any significant response to the historical temp (Fig 2), rather a swift recovery from the 1902-low, which would be difficult to attribute to Santa Maria since it happened late in October (arguably after the growing season in 1902). Should perhaps be a bit more cautious in attributing any response in the tree-rings to this event.

Figure 2 Caption: L567 not clear what the difference is between circles triangles and crosses. I guess circles are coincidence with known volcanic eruptions?

Figure 2 It is a bit strange that the authors do not extend the analysis of the volcanic cooling back to 800CE. I say this because the 3P-STREC clearly does not add clarity to volcanic cooling episodes, rather the opposite. If there would be an elegant way of using the D-STREC for the volcanic inferences I would not mind this. The 3P-STREC is of course important for 4P-STREC to constrain the more persistent developments of the temperature history and should be kept in parallel.

L129-131 Just to statistically consolidate the results it would be great to have a test and a p-value of how significant these results are. McCarroll et al., 2015 Holocene, designed an extreme value capturing test that I think the authors could be inspired by.

They write “The probability of capturing a given number of extreme years by chance can be calculated using the binomial distribution, providing a simple non-parametric test of statistical significance.”

Alternatively you could randomly sample 32 years and check which of those that are associated with volcanisms, and repeat that X times to calculate significance using percentiles. I am guessing the results to be highly significant, and this could be used to show this. Perhaps even the 3 NHET number of coincident volcanic events are a significant amount, but this is more uncertain..

...Would be a good addition to strengthen the conclusion that volcanism drive a substantial proportion of temperature variability in this region.

Table S9 I understand the rationale for choosing RSFi if it produces a better fit among chronologies and with the climate target. However, a standardization/detrending using RCS type methods will only be able to affect mid- to low-frequencies differently. Therefore it would be easier to identify the method of choice when smoothing the time-series with let's say a 10-year spline filter prior to correlation. This could complement table S9, that should be kept.

Reply to Reviewer Comments

Reply to Reviewer #1:

Overview

This study by Wang et al. develops new MXD chronologies and reconstructs the last millennium summer temperature of the Northeastern North America (NENA) region. Their new reconstruction shows stronger and less delayed volcanic responses than an existing reconstruction based on ring width and cellulose, confirming the superiority of MXD as a recorder of high frequency temperature variation. Then, they combine these two reconstructions to form a new one for NENA that shows overall higher skill, and find a good agreement with climate model simulated Northern Hemisphere extratropical (NHET) summer temperature regarding temporal variation. The comparison to single-forcing simulations indicates that tropical volcanic eruptions are the main forcing that synchronizes the NENA and hemispheric summer temperatures.

In my opinion, these results are of importance and interest to the community, and can stimulate further studies. The manuscript is overall in good quality, with a clear structure, and analyses being thorough and to the point. I have only a few minor comments that I list below. Once those have been addressed, I recommend the work be accepted for publication.

We thank the reviewer for his/her positive comments on our manuscript. Point-by-point responses are listed below.

Details

Regarding Figure 4b:

+ This figure shows the comparison of the reconstruction 4P-STREC and the model simulated summer temperature over land 30-90N, using a 20-yr low-pass filter. What is the purpose of applying a low-pass filter here, and why 20-yr? Explanations are needed in the main text.

The purpose of using a low-pass filter is to highlight the multidecadal variability of the temperature series and its synchronization with the northern hemisphere. To clarify our choice of a 20-yr filter, in the caption of Figure 4 we added “all the time series are smoothed using a 20-year low-pass Butterworth filter to highlight multidecadal and long-term variations” (Lines 656–658). We also considered the reviewer's suggestion and added a 20–100-yr bandpass filter to show correlations when focusing only on the “multidecadal variability”. These results are displayed in Figure 4c–e and Supplementary Table 8.

+ Out of curiosity, what is the comparison like between the reconstruction 4P-STREC and the model simulated NENA summer temperature? A better consistency is expected, but it could still be a useful apples-to-apples comparison to be displayed in the supplementary information.

In fact, we had compared the 4P-STREC reconstruction with simulated air temperatures averaged over the study region (not shown in the previous manuscript). The correlations were not better than those computed between 4P-STREC and NHET simulations, likely because general climate models have poorer performance in estimating regional than larger-scale climate and simulations of regional climate are more strongly associated with modeled internal variability than hemispheric simulations. In the revised manuscript we added these correlations in Supplementary Table 8.

+ What is the comparison like between the model simulated NENA and NHET summer temperatures? If the NENA-NHET synchronization exists in model simulations, it could also support the main message of this study.

We followed the suggestion to compare simulated regional and NHET temperatures. Correlation coefficients can be found in the Supplementary Table 8 of the revised manuscript. We also added the following sentence to emphasize this result in the main text: “This is also well supported by strong NENA-NHET coherence in individual CMIP5 simulations (Supplementary Table 8)” (Lines 192–193).

Reply to Reviewer #2:

This paper presents a new tree-ring maximum latewood density (MXD) series from northeastern North America covering the last millennium. Such data is very, very welcome and quite exciting, as this region is still a data-gap in terms of high signal-to-noise proxy data for Common Era temperature reconstructions (as the authors accurately point out). These data will be extremely valuable for enhancing large-scale temperature reconstructions. The analysis here is interesting in several ways, particularly in how it extends, modifies meaningfully, and enhances previous analyses using non-MXD records from the same trees, species, and region. All the methods seems reasonable to me - mostly they are standard dendrochronological methods and where necessary certain innovations are used to deal with the challenges of this particular species and collection. But these all seem reasonable. I do always have some concerns about the use of Signal Free detrending, in that it can definitely cause (ironically) end-effect biases under certain circumstances and for certain datasets. However, I do not see any obvious evidence for the existence of these biases in this case.

We are grateful to the reviewer for acknowledging the importance of our research, and his/her very constructive and detailed comments. We have taken this opportunity to revise the manuscript based on these suggestions. Point-by-point responses are listed below.

My major concerns are about the strength of claims about the insignificance of internal variability and how this is supported (or not) by the data and analyses shown in the manuscript. I don't see any reason to think these concerns would be ultimately disqualifying for the manuscript, but I would like the authors to consider my substantive and minor comments below and to reconsider the strength of some of their claims or statements about internal vs forced variability.

We have considered the reviewer's concerns about the strength of internal variability at the regional scale. In the revised version, instead of saying that internal variability is relatively weak, we now place more emphasis on the strength of volcanic eruptions in shaping regional and NH multidecadal temperature variability. Explanations and changes are detailed under Point 3 below.

1. I'm curious about the motivation of combining the new MXD data presented here with the weaker temperature proxies in 3P-STREC (e.g. Line 98 and 99). What utility is there in adding less sensitive proxy data to a much improved MXD temperature reconstruction, especially given the complex mechanistic relationships between isotopes and temperature? The authors (Line 93 and Line 101) claim the the four proxy reconstruction is better based on correlation values, but the difference is only $r=0.72$ vs. $r=0.77$, and it is unlikely that the magnitude of difference between these two values is actually significant (which can be measured with a Z test), particularly since the gain comes while adding additional series (which would inflate the variance explained simply because of the additional predictors). Why not simply use the D-STREC reconstruction (and then continue as in various places in the supplemental material to compare/contrast with the earlier 3P-STREC and STREC reconstructions from Gennaretti?). In particular, it would be useful to compare the similarities and highlight the differences between the tree-ring STREC reconstruction, the multiproxy/isotope 3P-STREC reconstruction, and the current D-STREC MXD reconstruction independently because each of these has been previously put forward as a regional temperature reconstruction and earlier conclusions about e.g. volcanic eruptions and climate cooling have been drawn from these as well. How do these three generations of Quebec black spruce reconstructions differ, particularly with respect to the impact of volcanic eruptions like Samalas etc., once outside of the calibration period? For instance (Line 125-126), the current reconstruction has Samalas cooling as relatively small, but the original STREC reconstruction from Gennaretti et al. highlighted cooling around the time of Samalas as one of the major results of that earlier paper a la Miller et al. Similarly, in Lines 132-134 the authors find an extended significant cooling following tropical eruptions using SEA, which is similar to mixed proxy reconstruction superposed epoch analysis but different from MXD-only reconstructions like that of Schneider et al. 2015. In Figure S7, the separation of D-STREC from 3P-STREC suggests the latter is biased in its recovery of the sharp cooling related to volcanic events.

4P-STREC and D-STREC reconstructions are very similar at all time scales. However, like reviewer 2 underlines it, D-STREC is a mono-proxy reconstruction. Its noise component is probably better constrained than that of the multi-proxy 4P-STREC reconstruction. Therefore, we retained the suggestion to use only the D-STREC reconstruction in the revised manuscript. As pointed out by the third reviewer, the volcanic signature does not benefit from the inclusion of additional proxies. Relevant figures and statistics have been updated accordingly throughout the revised manuscript.

We have not added a comparison of reconstructed temperatures between D-STREC and the earlier STREC reconstruction. The earlier STREC was developed from tree-ring widths (TRW) by combining subfossil and living-tree stems sampled from varying heights on the trees. In a subsequent paper, we showed (Autin et al. 2015; Dendrochonologia) that such TRW datasets are prone to biases at the recent chronology end if they are standardized with common RCS

techniques. Consequently, STREC is negatively biased during the calibration period over the 20th and 21th centuries. This problem was corrected when developing the 3P-STREC reconstruction (Gennaretti et al., 2017 Climate Dynamics), including a comparison between STREC and 3P-STREC. To avoid repeating this comparison, here we only compare the temperature reconstructions of D-STREC and 3P-STREC (revised Fig. 2c). In the Methods section of the revised manuscript, we created a new paragraph heading (Comparison with the 3P-STREC temperature reconstruction) to explain how 3P-STREC was recalibrated, using the same temperature target as D-STREC, to allow direct comparison with D-STREC (see Lines 264–277 in the Methods section).

However, to demonstrate the much-improved temperature sensitivity of D-STREC versus previous reconstructions, we compared the spatial domains of significant correlations between proxies and instrumental temperatures (revised Fig. 1 b-e), as well as various calibration statistics (revised Supplementary Table 3) for D-STREC, 3P-STREC and the corrected TRW component of 3P-STREC. We modified this sentence in the main manuscript: “Compared to both 3P-STREC and its tree-ring width component (Methods), the D-STREC reconstruction skill is substantially improved (Supplementary Table 3), with a much larger spatial domain (Fig. 1) and better constrained uncertainties (Fig. 2c)” (Lines 89–91).

In addition, as suggested, we added reference to the STREC reconstruction when discussing the Samalas eruption at Lines 132–136.

2. On volcanic SEA: Figure 3: The claim in the manuscript is that the cooling post-tropical eruption last for a decade, but in Figure 3 the SEA composite crossed back into 'non significant' after just a few years. In Figure S6 and S12, SEA typically involves a resampling (random with replacement or block) to assess significance levels, but that doesn't seem to have been done here as it was in Figure 3. Would the use of conventional SEA confidence intervals change the inference about the length of cooling in the supplemental figures and in the manuscript? This question is even more important since Figure 3 and Figure S7 does have these types of confidence limits and both suggest a return to within the baseline of cooling by 3 to 5 years other thereabouts.

Our claim is based on the fact that significant cooling frequently occurs during the ten-year time period after tropical eruptions (Figure 3a). In the revised manuscript, we describe the post-eruption cooling more rigorously (Lines 146–148): “tropical eruptions caused about a 5–10 years of significant cooling (0.95 significance level) followed by an additional ~2-year recovery to the pre-eruption level” We also added that (Lines 152–154): “Comparisons of multiple subsets of volcanic events validate the more persistent cooling following tropical eruptions (Fig. 3a, b; Supplementary Fig. 5), a result not affected by closely spaced eruptions”

Concerning the display of the intervals for statistical significance in the various SEA figures, we applied the same blocking significance test for all SEAs (Lines 371–373 of the Methods section), except for the SEA on CRU data (Figure S12e, f in the original manuscript vs Supplementary Fig. 5 in the revised one) because only 2-3 events were available in this case. We have clarified this in the caption of the revised Supplementary Figure 5. Although the same method is used everywhere, Supplementary Fig. 6 (same Fig. number in the original and revised versions) displays the statistical significance in a somewhat different way because many SEAs are

compared simultaneously. Rather than displaying significance intervals for each separate composite, we display the percent of composites that are significant for each temporal lag. In addition, in the Fig. 3 caption, we clarified at Lines 645–646 that the significance intervals are based on the “all” events group: “Horizontal dashed Lines in a and b represent the 0.95 statistical significance level for the “All” group (Methods)”.

Figure S3 also shows that for the very large eruption of Tambora the reconstruction returns to baseline within 3 years. I'm concerned that something about the SEA (probably the event list and/or the significance testing) seems to suggest a longer recovery from a single event than is warranted. This not to say that multiple events (e.g. in the 1450s) couldn't combine to cause a ~decade of cooling, but rather that the recovery from a single event needs to be looked at more closely, particularly if the single large eruptive event in 1815 shows a rapid recovery (including in the historical data shown in Figure S3). And in Figure 3, Figures S11, S12, and Table S7: The event lists here contain several closely spaced eruptions, which could bias the SEA since there are actually multiple events in the post event window - how would the SEA (e.g in Figure 3) change if only the very largest events (e.g. Tambora or Pinatubo or larger) with at least 10 years of spacing were considered?

Cooling of a single event could be obscured by other forcings and internal climate variability (Fischer et al., 2007). That being said, a rapid recovery following Tambora may not fully reveal the average response of the climate system to tropical eruptions. By emphasizing the average response to several events and smoothing out the non-volcanic variations, SEA is more likely to show a comprehensive picture of volcanic response.

In the Supplementary Fig. 5c, d of the revised manuscript (previously Figure S12 c, d in the original manuscript) we consider only events ($SAOD_{NHET} \geq 0.03$) with at least 9 years of spacing. Results are generally in line with Fig. 3a, b, in terms of cooling persistence. SEAs on different subsets are similar to SEA using the full list, crossing back to zero at > 10 years after tropical eruptions. Consequently, we have made some modifications in the main text. At Lines 146–147, we replaced “On average, tropical eruptions caused about a decade of significant cooling” with “On average, tropical eruptions caused about 5-10 years of significant cooling”. In addition, at Lines 152–154, we added: “Comparisons of multiple subsets of volcanic events validate the more persistent cooling following tropical eruptions (Fig. 3a, b; Supplementary Fig. 5), a result not affected by closely spaced eruptions”.

3. Excessively strong claims about the importance of forced vs. unforced (decadal) variability. The authors based their claims that there is little unforced decadal variability at regional scales on Figure 4. Central to this claim is the correlations between low pass 20-year smoothed Northern Hemisphere reconstructions. The authors claim this provides evidence that at decadal scales their regional reconstructions matches large-scale reconstructions. However, there are reasons to approach this with caution: (1) as noted below, smoothing the series with a low-pass filter will enhance the correlation but reduce the degrees of freedom and this requires a modification of the significance level (or, more likely, some sort of simulation a la Ebisuzaki 1997) to assess a more accurate null; (2) Nearly all of the reconstructions considered use the STREC tree-ring reconstruction (NTREND, Stoffel and Guillet) - so the regional reconstruction includes information that is in the large-scale reconstructions. The lowest correlation between

4P-STREC and an existing reconstruction is somewhat worryingly with the Schneider et al. 2015 reconstruction, which is the only one to use pure MXD besides NTREND_D. Moreover, reconstructions including NTREND use an MXD only chronology from Quebec back to the 14th century. So, there is some degree of overlap in the fact that the large-scale reconstructions contain regional information that is likely reflected in the 4P-STREC reconstruction; (3) It is clear that while there is strong synchronicity between the regional reconstruction and the large-scale reconstruction during certain periods associated with multiple volcanic eruptions and widespread cooling (1450s, 1600, early 1800) there is disagreement at other decadal periods, particularly prior to 1300 CE (as highlighted by e.g. Esper et al. 2018 in *Dendrochronologia*). So, there is indeed support here (and elsewhere) for the idea that decadal scale cooling is synchronous for several periods of volcanic activity (again, particularly for periods during the LIA, partial coincident with solar minima, and specifically around 1450s, 1600, and early 1800s), but it is simply too strong to use the evidence here to downplay the role for internal climate system variability in influencing the magnitude of cooling and spatial patterns of cooling.

I would be happy for the authors therefore to suggest that volcanic eruptions synchronize decadal scale cooling, but the paper does not support the idea that internal climate system isn't important across a range of timescales and even during cold periods. In fact, one can see that (accounting for poor data coverage in many locations) in the Neukom et al. paper on non/synchronous warm and cold periods that what is different is not so much that places aren't cooler during periods of explosive volcanism, but rather the *_degree_* of cooling is different - with some places showing e.g. the 1600 or 1450s as coldest, and other regions showing different periods a relatively the coldest (this is a problem with the Neukom analysis, in addition to issues of data quality in those spatial reconstructions). Taking a closer look at NTREND, where data quality isn't as much of an issue (although data availability still is) one can see that the coldest years or decades are all associated with volcanism, but which period is the COLDEST varies in space (perhaps because of noise in the proxies, but also likely because of the influence of the combined nature of the forced plus unforced variability). Thus the claims in e.g. Lines 175 to 180 and the central framework of the paper (e.g. that internal climate system isn't important for decadal patterns of variability) are overstated.

We appreciate the reviewer's comments on our claim of the relatively weak role of internal climate variability and we agree that it was to some extent overstated in the previous version. Here are our responses to the three major points raised by the reviewer:

- 1. Reduced degrees of freedom: all the P values related to Pearson's r in this manuscript are calculated based on the method of PAGES 2k (2019), which compares correlation of real climate time series with random noises that share the same autocorrelation properties with the real data. See the "Correlation analysis and significance test" paragraph in the Methods section. Consequently, our P values are not affected by autocorrelation. All correlations between D-STREC and the NHET reconstructions and multi-model mean are found to be significant ($P < 0.01$) when using this method.*
- 2. Geographical overlap: the fact that the Quex and STREC datasets are included in many of the NHET reconstructions is probably not a significant cause of coherence between D-STREC and those NHET reconstructions. First, our study is now based on D-STREC, which shares much less data with the considered NHET reconstructions compared to 4P-*

STREC. The only data in common are those of the Quex dataset, which is a minor component of both D-STREC and the NHET reconstructions. Second, in the revised manuscript we consider an additional NHET reconstruction composite (Anchukaitis et al., 2017) from which we could exclude the gridded data covering the NENA spatial domain (see Supplementary Fig. 8 and Table 8 in the revised manuscript). This NENA-excluded time series is correlated with D-STREC at 0.47 during 851–2000 CE, a value higher than the correlation of D-STREC with Guillet et al., 2020, and even higher than the correlation with the whole NHET composite of Anchukaitis et al., 2017. This confirms that the "data overlap" issue is not significant. This new NENA-excluded NHET temperature series is introduced at Lines 170–171 in the main text.

- 3. The role of volcanic forcing: we replaced the claim that internal variability is not so important by a more conservative one centered on the relative importance of tropical volcanoes compared to all other potential drivers of multidecadal temperature variability. We designed a new analysis to support this statement. We excluded post-eruption years vs. random years in the correlation analysis. Results (Fig. 4d, e) confirm that tropical volcanoes are the most dominant driver of the region-NHET coherence at multidecadal timescale. Excluding years following tropical eruptions largely decreases this coherence, indicating that all other drivers have a low combined impact on multidecadal coherence compared to tropical eruptions. Accordingly, in the abstract (Lines 34–34), and at the end of the discussion (Lines 200–201) we replaced the claim that “our study downgrades oscillatory climate variability” with a claim that “our study highlights the the dominant role of tropical volcanism in shaping multidecadal temperature variations across spatial scales”.*

Additional comments

Line 71: remove 'unprecedented'

We replaced 'unprecedented' with “a large number of MXD...” in Line 65.

Line 81, 87, Figure 1: Do these statistics account for the influence of trends in the datasets (which tends to inflate both the effect size as well as the regional area of significance)? How does the effect size (r) and significance level change if only interannual variability is considered (high pass filtered)? Based on the global pattern of correlation, I suspect the field correlations shown in Figure 1 include the large-scale warming trend, which will bias these field correlations toward the appearance of large-scale coherency (e.g. see Figures 4 and 5 in Anchukaitis et al. 2017 in QSR).

In the original manuscript, these statistics were based on raw series with long-term warming trends. In the revised version of Fig. 1 correlation maps are based on 30-yr high-pass series. The significance test considers autocorrelation, as described in the Methods section. All the maps in Fig. 1 do not differ substantially from those in the previous version.

Figure 1: How is the inset map in Figure 1a created? Presumably not all of these sites were

correlated against the field individually - so is this the average of the MXD sites shown? As above, have you accounted for autocorrelation and trend and its effects on these types of field correlations?

The inset map was created by taking the maximal positive correlation value across the individual spatial correlation fields of the 12 MXD series. In the revised version correlations were calculated using 30-year high-pass series. We now better explain how the inset map was created in the caption of Fig. 1. See Lines 610–611.

Line 101: '96% of the CRU observational target constrained within its 95% confidence interval' - but wouldn't you expect this by definition (e.g. that the confidence interval at a certain level contains the target data at that level)? I'm unclear why this is particularly diagnostic, at least as described.

We removed this sentence.

Line 114, 115: '~1251–1930 CE' - this is an unusual definition of the LIA, particularly the idea of extending it to the 1900s! - why not use the IPCC definition of 1450 to 1850 CE? I see this reconstruction has anomalous cooling into the 1900s indeed, but that doesn't mean a quasi-global definition of the LIA shouldn't be applied.

We now use the definition of the MCA and LIA according to IPCC AR5, and describe the region-specific features (Lines 107–116).

Line 121: there is some doubt if this Kuwae - see e.g. Hartman et al. 2019 and the discussion and citations therein

We have considered the 1457 Kuwae? as an unknown eruption in the text (Line 126) and Supplementary Table 7 of the revised manuscript.

Line 127-128: Perhaps, maybe, but there are lots of other uncertainties about Samalas that should be mentioned here related to the volcanic forcing - e.g. see for instance at least Timmreck et al. 2009 in GRL, Marshall et al. 2019 in JGR, Wade et al. 2020 recently in PNAS, and specifically with regarding ENSO and Guillet's hypothesis, the papers by Dee et al. in Science in 2020 and Zhu et al. in Nature Communications in the last few months (which do not support an El Nino event at that time)

We agree and have modified the corresponding sentence accordingly. We also added citations to the relevant studies of Timmreck et al., Marshall et al. and Wade et al.: "...This moderate response is in line with the 3P-STREC reconstruction (Fig. 2c) and is probably due to region-specific volcanic responses²⁵ and complex atmosphere chemistry" (Lines 130–132).

Line 153: Does the significance level here account for the autocorrelation and loss of degrees of freedom induced by the smoothing? I think so, based on Lines 292-304, but I am somewhat surprised at this level of significance, given the loss of degrees of freedom

Yes, all the significance levels of correlations in this manuscript are based on the method of PAGES2k (2019), which compares correlations between two timeseries with random noises (Lines 308–319).

Line 167: omit 'relatively high' - could substitute with 'significant' (assuming as per my comment above accounting for the loss of degrees of freedom still leaves these relationships significant)

We replaced "relatively high" with "significant" (Line 189).

Line 175: But Mann et al. is specifically talking about the AMO, which is difficult to separate - and indeed projects onto - the large-scale Northern Hemisphere mean. This study, which is specifically about summer air temperature in a particular region compared to other large-scale reconstructions, says nothing in particular about the dynamics driving the AMO. This claim here can therefore be omitted, since it is not directly relevant.

We omitted the citation to Mann et al. here.

Line 308: Why use VEI as a criteria, when VEI does not correspond well to climatic forcing (which is related to SAOD)?

We agree with the reviewer. VEI from GVP does not directly correspond to climatic forcing. In fact, very few events were matched to eruptions based only on VEI in the previous manuscript. In the revised version, we used VEI only to inform on eruptions: “Locations of eruptions were identified from the corresponding volcanoes provided by the confirmed eruption list (dating uncertainties < 1 year) of Global Volcanism Program⁶¹ (GVP; tropical: 30 °S–30 °N, NHET: 30–90 °N) or from the ice-core datasets (for unidentified events)” at Lines 322–325.

Table S1: What would a similar analysis to Figure 4 look like with the individual MXD records in this Table? I suspect that most of these show the major cooling events around the 1450s, 1600, and early 1800s, since these appear across large-scale reconstructions and in spatial reconstructions like NTREND and Guillet, but this would provide a way to look at the ratio of common decadal-scale forcing between widely separated series not sharing any data or even regional similarities

We have tested correlations between individual records in Table S1 and D-STREC. Not all correlations are as strong as between D-STREC and NHET reconstructions, although a few individual MXD records show high coherence with NHET simulations. This very likely suggests that data quality is variable (Esper et al., 2016; QSR) and plays an important role in the strength of these correlations. We prefer to focus on time series that cover a large spatial domain across NHET because we want to emphasize that D-STREC behaves like the best known NHET tree ring-based reconstructions, even if D-STREC is regional in nature. To eliminate the potential issue of data overlap we considered one additional gridded NHET reconstruction composite (Anchukaitis et al., 2017) from which we excluded the gridded data covering the NENA spatial domain corresponding to our reconstruction target (Supplementary Figure 8). Correlation between D-STREC and this new series (labelled as “NTREND-excl” in the revised Fig. 4) was

even stronger than the correlation between D-STREC and the full NHET mean of Anchukaitis et al., 2017, indicating that the data overlap issue is negligible.

Figure 4: What if instead of a low-pass filter you used a band-pass to isolate 20 to (for instance) 40 year periods (and remove both the high and low frequencies)? Does this give the same impression as the statistics here, or is centennial-scale variability (left in place by the low-pass filter) influencing the correlations at all here?

As expected, correlations between band-pass filtered series (20–100 year) are lower than those between 20-year low-pass series, as longer-term trends are filtered out. However, these correlations between D-STREC and multi-model mean are still significant, and similar to the level of the correlation between D-STREC and volcano-only simulations (Supplementary Fig. 9g). Therefore, these weaker correlations do not alter our final conclusion (Lines 182–183). To be more transparent, we added these results in Fig. 4c and Supplementary Table 8.

Reply to Reviewer #3:

Reviewer #3 (Remarks to the Author):

Comments to Wang et al “Tropical volcanoes synchronize eastern Canada with Northern Hemisphere millennial temperature variability”

The manuscript presents an exceptionally well-replicated regional network of maximum latewood density data from eastern Canada. The region is somewhat underrepresented in hemispheric millennium long temperature reconstructions 1) because so far only ring width comfortably includes MCA and this proxy has documented weaknesses compared to MXD, 2) because MXD previously sampled in the region only reaches back to mid-1300 CE with modest replication, 3) because stable isotopic tree-ring data was never produced on an annual scale, which severely challenges any analyze of volcanic signatures. The new data is in itself a substantial contribution to the field of late Holocene paleoclimatology. The new temperature reconstruction which has substantial advantages over previous attempts (chronology length, signal strength, reduced confidence intervals due to the large replication and varied spatial representation and proxy type) is ideally suited to explore volcanic cooling and its overall effects on temperature variations.

We appreciated that the reviewer found our work valuable. His/her comments, in particular the suggestions of some statistical tests, were very helpful. Below we provide point-by-point responses to these comments.

The authors find that the variability of the reconstruction is largely driven by volcanism, at least prior to when GHG started to become a major forcing agent. Even comparable to the volcanically induced variability in NH products from either tree-rings or climate models. The amount of volcanic cooling that cuts through the stochastic nature of internally forced climate is remarkable. However, it is also not completely surprising since the included sites from the region overall express a quite low inter-series correlation (\bar{r}) FigS2, which means that common

features such as volcanic cooling will be high-graded, similarly to what is observed for NH tree-ring based reconstructions and climate models. The reconstruction thus serves as very valuable regional constraint of volcanic cooling forcing for climate models. I particularly think of Samalas 1257 which only modestly impact the temperature here. Here the models and the 4P-STREC differ substantially.

We added some texts in the revised manuscript to emphasize the D-STREC vs. simulations difference concerning the Samalas, along with citations of some previous proxy studies that highlighted this phenomenon (Lines 183–187): “The most prominent difference between D-STREC and simulations during the last millennium concerns the impact of the Samalas eruption (Fig. 4b). D-STREC provides complementary proxy evidence that the cooling effect of the Samalas was disproportionately low compared to its amplitude in CMIP5 climate model simulations^{8, 29, 37”}.

The manuscript is well written, carefully worked through with overall sound methodology. My main concern regards the inclusion of 3P-STREC. I understand the rationale because the T signal is strengthened in the lower frequencies by introducing more data from different sources, but the volcanic signature does not benefit from its inclusion and this is the main part of the manuscript. The authors try to tackle this problem by decomposing the signals in low and high frequency components but I worry that this will anyway smear the possibly more distinct signature. Moreover, the author could potentially extend the volcanic analysis to cover >200 yrs more when using only the D-STREC. I guess I am trying to say that it would be better if the T-reconstruction would be done using 4P-STREC, and the volcanic analysis in its full extent relied more heavily on only D-STREC.

The purpose of combining 3P-STREC with D-STREC was to make our final reconstruction product more robust in the domain of low frequencies. However, we agree better calibration statistics do not guarantee that the 4P-STREC reconstruction is robust along its entire length, particularly given that 3P-STREC has lagged responses to volcanic eruptions. Therefore, considering also that the second reviewer made a similar comment, we only kept the D-STREC reconstruction in the revised manuscript. Relevant figures and statistics have been updated accordingly. However, we did not extend the SEA analysis before 1000 CE due to increasing dating uncertainty of ice core records (we’ve shown that in Lines 140–144) and very few earlier events meeting our criteria.

It would also be more transparent to present temperature calibrations for the full D-STREC overlap with T-data, that is, up to 2017.

In the original manuscript we used 1905–2006 CE as the full calibration period because we wanted to combine all proxies into the 4P-STREC reconstruction. The common interval of the three new MXD records (L135, L20, L105) ends in 2008 while 3P-STREC ends in 2006. Calibrating over a common period provides equal weights for all local temperature series. Although in the revised manuscript we no longer consider the 4P-STREC, we have kept the 1905–2006 calibration for D-STREC because lengthening the calibration by 2 years to 2008 would have implied running a computer for 30 days to recompute the D-STREC reconstruction

plus the 100 members needed for evaluating the proxy-level uncertainty in the Bayesian framework.

In the Methods section of the revised manuscript, we nevertheless provide the r value during 1901–2017 CE ($r=0.76$) along with the calibration r (1905–2006 CE; $r= 0.75$) to be more transparent (Lines 247–249).

Below I list some minor comments:

L63-68 Perhaps refrain from revealing the conclusions in the introduction. The passage could be rephrased to “Based on this dataset, we test the proportion of forced and stochastic variability... on the regional reconstruction... and compare to that of NH hemispheric...”

The guideline of the journal indicates: “Introduction-The results of the current study must only be discussed in this final paragraph.” We now rephrase the sentence: “Based on this dataset, we show that an important proportion of the multidecadal temperature variability in NENA is externally forced by explosive volcanism, especially tropical eruptions, synchronizing regional variations with those of the whole Northern Hemisphere” to emphasize a bit the results without focusing too much on the conclusions (Lines 59–62).

L81 Why was the instrumental MJJA only presented up to 2006? Also in Fig S9. Is the replication of the reconstruction fading after this or is it something with the instrumental data?

Same as above. We have not set 1901–2017 as the full calibration period due to the varying lengths of the MXD records included in D-STREC. Yes, the replication of the reconstruction fades after 2008 CE. In the revised manuscript, we show the full length (up to 2017 for CRU and 2010 for the historical record) of instrumental temperatures in the revised Fig. 2d.

L94 Should mention that the negative spike in $t+1$ is nicely reproduced by the reconstruction but $t+2$ appears to exhibit a memory from $t+1$, and that the recovery to track instrumental T only occurs in $t+3$. That is, there may be some memory effect (Esper et al., 2015) also in the D-STREC and 4P-STREC? Whereas the 3P-STREC clearly has a muted cooling but also a lagged recovery..

We added the following text: “...although the recovery is one year longer for sites farther north (Supplementary Fig. 3). We speculate that this phenomenon reflects the increasing severity of post-eruption growth stress towards the northern treeline.” (Lines 95–97).

114 not necessary to present the abbreviation of an expression, remove

We used “with respect to” everywhere in the revised manuscript.

L114 This is a pretty extended definition of the LIA. Present what the general definition of the LIA is with a reference, and then you can go into the regional expression from your reconstruction.

In the revised manuscript we use the definition of MCA and LIA by IPCC AR5 (Lines 111–116).

Table S3 I note that although you split the calibration period into two periods, I cannot find any statistical tests for validating the calibration, and thus reconstruction. Please include statistics such as Reduction of error (RE) and coefficient of efficiency (CE) as a minimum, but also consider using the stationarity test discussed in Wilmking et al., 2020 GCB.

The Bayesian framework differs from regression-based methods as it provides probabilistic estimation of past climate. Reduction of error (RE) and coefficient of efficiency (CE) are considered inadequate to assess ensemble-based Bayesian reconstructions (Gneiting and Raftery, 2007; Werner et al., 2018), while continuously ranked probability score (CRPSpot) and the reliability score (Reli) are more suitable (smaller values indicate better skill). In the original manuscript, we presented these two scores in Supplementary Table 3, but only for reconstructions developed using the full calibration period. In the revised manuscript (Supplementary Table 3), we also use a split-period calibration/validation approach (1905–1955 and 1956–2006) to validate the D-STREC and 3P-STREC reconstructions.

The BTFS stationary test used by Wilmking et al., 2020 was developed by Buras et al., 2017. This is a regression-based approach that iteratively calculates linear model parameters in two sub-intervals using bootstrap resampling with replacement. Although in the original work authors did not specify how many data should be sampled, one can expect that the length of sampled data should be the same as the length of each sub-interval. Using a number of 51 (half length of the 1905–2006 period) we obtained a significant result. However, in the R code provided by the authors, 2N data were sampled from N available data points (N is the half length of the full calibration period) in each sub-interval (over-sampled data might induce a bias). As described by Buras et al., 2017, BTFS performance is significantly affected by sample size. Indeed, we found that results varied from significant to insignificant by changing N from 51 to 102. We thus think this method is not suitable to be used here. The more widely used split-period calibration/validation approach should be powerful enough.

L121 See Esper et al., 2017 Bulletin of Volcanology, for a better date for Kuwae

We acknowledge that the 1450s eruptions are still under debate, thus we only mentioned the years of these eruptions without attributing them to a particular volcano in the main text (Line 126) and Supplementary Table 6.

L122-123 The Santa Maria eruption did not seem to produce any significant response to the historical temp (Fig 2), rather a swift recovery from the 1902-low, which would be difficult to attribute to Santa Maria since it happened late in October (arguably after the growing season in 1902). Should perhaps be a bit more cautious in attributing any response in the tree-rings to this event.

Yes, we agree and this comment is even more pertinent regarding the 4P-STREC reconstruction, so Santa Maria is no longer discussed in the revised manuscript.

Figure 2 Caption: L567 not clear what the difference is between circles triangles and crosses. I guess circles are coincidence with known volcanic eruptions?

We added a legend inside Fig. 2b.

Figure 2 It is a bit strange that the authors do not extend the analysis of the volcanic cooling back to 800CE. I say this because the 3P-STREC clearly does not add clarity to volcanic cooling episodes, rather the opposite. If there would be an elegant way of using the D-STREC for the volcanic inferences I would not mind this. The 3P-STREC is of course important for 4P-STREC to constrain the more persistent developments of the temperature history and should be kept in parallel.

As explained above, there are very few precisely dated eruptions before 1000 CE for SEA analysis. Extending the SEA to the first millennium could introduce biases due to dating uncertainty. For example, we performed a binomial distribution test (suggested by the reviewer in the next comment) to understand whether the attribution of coldest years to large eruptions is significant. The attribution for the 772–2017 period is less significant than that for the 1000–2017 period ($p = 0.051$ vs. 0.021), suggesting that volcanic dates from ice cores are likely less accurate in the first millennium. We thus kept the same SEA results.

L129-131 Just to statistically consolidate the results it would be great to have a test and a p-value of how significant these results are. McCarroll et al., 2015 Holocene, designed an extreme value capturing test that I think the authors could be inspired by.

They write “The probability of capturing a given number of extreme years by chance can be calculated using the binomial distribution, providing a simple non-parametric test of statistical significance.”

Alternatively you could randomly sample 32 years and check which of those that are associated with volcanisms, and repeat that X times to calculate significance using percentiles. I am guessing the results to be highly significant, and this could be used to show this. Perhaps even the 3 NHET number of coincident volcanic events are a significant amount, but this is more uncertain....Would be a good addition to strengthen the conclusion that volcanism drive a substantial proportion of temperature variability in this region.

Thank you for this helpful suggestion. We have used a binormal distribution to test the significance of the attribution, which is significantly different from a random sampling (See the Methods section, Lines 337–340).

Table S9 I understand the rationale for choosing RSFi if it produces a better fit among chronologies and with the climate target. However, a standardization/detrending using RCS type methods will only be able to affect mid- to low-frequencies differently. Therefore it would be

easier to identify the method of choice when smoothing the time-series with let's say a 10-year spline filter prior to correlation. This could complement table S9, that should be kept.

We have added correlations based on 10-year low-pass filtered time series in this Table (which is Supplementary Table 10 in the revised manuscript).

REVIEWERS' COMMENTS

Reviewer #2 (Remarks to the Author):

The authors have done an excellent job addressing my comments, particularly on 'synchronization' of multidecadal temperature anomalies, addressing SEA questions, and emphasizing the power of the MXD (D-STREC) proxy series. After reading their response and the new manuscript with tracked changes, I find that they have addressed my major comments to my satisfaction and I believe the manuscript is ready to be accepted.

Reviewer #3 (Remarks to the Author):

Dear Authors, I consider the revisions made to be sufficient to accept the manuscript in the current form. Congratulations to an exciting piece of work!

Reply to Reviewer Comments

Reply to Reviewer #2:

Reviewer #2 (Remarks to the Author):

The authors have done an excellent job addressing my comments, particularly on 'synchronization' of multidecadal temperature anomalies, addressing SEA questions, and emphasizing the power of the MXD (D-STREC) proxy series. After reading their response and the new manuscript with tracked changes, I find that they have addressed my major comments to my satisfaction and I believe the manuscript is ready to be accepted.

We thank the reviewer for his/her comments on our manuscript.

Reply to Reviewer #3:

Reviewer #3 (Remarks to the Author):

Dear Authors, I consider the revisions made to be sufficient to accept the manuscript in the current form. Congratulations to an exciting piece of work!

We thank the reviewer for a second round of review.